# Uncomputability of phase diagrams

Johannes Bausch [1✉], Toby S. Cubitt[2✉] & James D. Watson [2✉]

The phase diagram of a material is of central importance in describing the properties and behaviour of a condensed matter system. In this work, we prove that the task of determining the phase diagram of a many-body Hamiltonian is in general uncomputable, by explicitly constructing a continuous one-parameter family of Hamiltonians $H(\varphi)$, where $\varphi \in \mathbb{R}$, for which this is the case. The $H(\varphi)$ are translationally-invariant, with nearest-neighbour couplings on a 2D spin lattice. As well as implying uncomputablity of phase diagrams, our result also proves that undecidability can hold for a set of positive measure of a Hamiltonian's parameter space, whereas previous results only implied undecidability on a zero measure set. This brings the spectral gap undecidability results a step closer to standard condensed matter problems, where one typically studies phase diagrams of many-body models as a function of one or more continuously varying real parameters, such as magnetic field strength or pressure.

[1] CQIF, DAMTP, University of Cambridge, Cambridge, UK. [2] Department of Computer Science, University College London, London, UK.
✉email: jkrb2@cam.ac.uk; t.cubitt@ucl.ac.uk; ucapjdj@ucl.ac.uk

Phase transitions and phase diagrams have been a central area of study in condensed matter physics for well over a century. In particular, in the second half of the twentieth century, interest in superconductors and topological phases spurred work on quantum phase transitions: a discontinuous change of a macroscopic observable happening at zero temperature due to the change in some non-thermal parameter[1].

The phase diagrams for many materials have been well studied both experimentally and theoretically. There exist numerous algorithms that are heuristically effective at computing properties of many-body quantum systems, such as the Density Matrix Renormalisation Group for one-dimensional (1D) gapped systems or density functional theory[2,3]. Classic toy models include the 1D transverse field Ising Model, which is known to have a transition from an unordered to ordered phase at a critical magnetic field strength[1]. Beyond that, materials with exotic phases, such as topological insulators, or the fractional quantum Hall effect are becoming increasingly important to understand as they become more applicable to real-world applications[4,5].

Yet, the quantum phase diagram for such systems can be highly complex. Numerical simulations of quantum systems are computationally difficult, and may even be intractable[6,7]. Experimentally and computationally one of the best studied is the 2D electron gas—a model for free electrons in semiconductors—which is well known to exhibit a complex phase behaviour: the system undergoes a large number of phase transitions, most notably those associated with the quantum Hall effect. Indeed, the phase diagrams of such systems are known to be incredibly rich, with some producing Hofstadter butterfly patterns with an infinite number of phases[8]. All of these are important instances of the general problem of computing the phase diagram of a Hamiltonian, which classifies the system's state with respect to a macroscopic observable (such as global magnetisation), and with respect to a parameter of the Hamiltonian (such as a transverse field strength).

Quantum phase transitions are associated with the spectral gap of the Hamiltonian closing. More precisely, a non-analytic change in the ground state energy is a necessary (although not always sufficient) condition for a phase transition to occur, and a closing spectral gap is necessary (although not always sufficient) for a non-analytic change in the ground state energy to occur. Cubitt, Perez-Garcia and Wolf[9,10] showed that given a (finite) description of a translationally invariant, nearest-neighbour Hamiltonian on a 2D square lattice, deciding whether it has a spectral gap or not is at least as hard as solving the HALTING PROBLEM. This was subsequently extended to the case of 1D Hamiltonians[11].

In this work, we prove that no general algorithm for determining the phase diagram of a system can exist, even given complete knowledge of the microscopic description of the system's interactions. To show this, we explicitly construct a continuous, one-parameter Hamiltonian $H(\varphi)$ on 2D lattice with a fixed, finite-dimensional local Hilbert space $\mathcal{H}_A \oplus \mathcal{H}_B$, for which determining whether the low energy subspace below some energy cut-off is supported entirely on the A or B subspace is undecidable (where it is guaranteed that one of the two cases holds on a set of positive measure in the parameter space of the model). With respect to the parameter $\varphi$, the phase diagram determined with respect to a macroscopic observable $O_{A/B}$ that measures support on $\mathcal{H}_A$ vs. $\mathcal{H}_B$ is thus uncomputable. This observable can also be restricted to a single lattice site, with the same conclusion.

## Results

The quantum many-body systems we will consider are translationally invariant, nearest-neighbour, 2D spin–lattice models. The $L \times L$ square lattice with open boundary conditions will be

denoted as $\Lambda(L)$; for brevity, we leave the lattice size implicit whenever it is clear from the context. Each lattice site is associated with a spin system with local Hilbert space of dimension $d$, $\mathbb{C}^d$. The spins are coupled with a nearest-neighbour, translationally invariant Hamiltonian with local terms $h^{col}, h^{row} \in \mathcal{B}(\mathbb{C}^d \otimes \mathbb{C}^d)$, such that $\max\{\|h^{row}\|, \|h^{col}\|\} \leq 2$. Since we are interested in phase transitions—identified by a discontinuous change of a macroscopic observable $O_{A/B}$, which strictly speaking can only occur in the thermodynamic limit of infinitely large lattices—we will take the thermodynamic limit by letting $L \to \infty$. An alternative definition of a quantum phase transition is a non-analytic change in the ground state energy[1]. This will also be satisfied with our construction. The resulting Hamiltonian over the entire lattice is then

$$H^{\Lambda(L)} := \sum_{i=1}^{L} \sum_{j=1}^{L-1} h^{row}_{(i,j),(i+1,j)} + \sum_{i=1}^{L-1} \sum_{j=1}^{L} h^{col}_{(i,j),(i,j+1)}. \quad (1)$$

As well as being distinguished by the observable $O_{A/B}$, the two phases are also distinguished by the spectral gap of the Hamiltonian $H^\Lambda$, defined as the difference between the smallest and second smallest eigenvalue of the Hamiltonian:

$$\Delta(H^{\Lambda(L)}) := \lambda_1(H^{\Lambda(L)}) - \lambda_{\min}(H^{\Lambda(L)}). \quad (2)$$

As in ref. [9], we then define a Hamiltonian to be *gapped* if there exist $\gamma$ and $L_0$ such that the spectral gap $\Delta(H^{\Lambda(L)}) \geq \gamma$ for all $L > L_0$; and *gapless* if the spectrum above the ground state becomes dense in an interval $[\lambda_{\min}(H^{\Lambda(L)}), \lambda_{\min}(H^{\Lambda(L)}) + c]$ for some $c > 0$ in the thermodynamic limit (see Supplementary Definitions 1.1 and 1.2 for mathematically rigorous statements). Throughout the paper, we will be using the notion of a *continuous family of Hamiltonians*, which—loosely speaking—is a family of Hamiltonians $\{H_i(\varphi)\}_{i \in I}$ such that each $H_i(\varphi) = \sum_j h_j(\varphi)$, and the matrix elements of $h_j(\varphi)$ depend continuously on $\varphi$ (see Supplementary Definition 1.3).

Our main result is an explicit construction of a one-parameter continuous family of Hamiltonians, such that for all values $\varphi \in \mathbb{R}$ of the external parameter, the system is guaranteed to be in one of two possible phases, distinguished by an order parameter given by the ground state expectation value of a translationally invariant macroscopic observable $O_{A/B}$. However, determining which phase the system is in is undecidable, hence the phase diagram of the system as a function of $\varphi$ is uncomputable. More precisely, we prove the following theorem:

**Theorem 2.1** (Phase Diagram Uncomputability) *For any given Turing Machine (TM), we can construct explicitly a dimension $d \in \mathbb{N}$, $d^2 \times d^2$ matrices $a, a', b, c, c'$ and a $d \times d$ matrix $m$ with the following properties:*

(i)   *$a, c$ and $m$ are diagonal with entries in $\mathbb{Z}$.*
(ii)  *$a'$ is Hermitian with entries in $\mathbb{Z} + \frac{1}{\sqrt{2}}\mathbb{Z}$.*
(iii) *$b$ has integer entries.*
(iv)  *$c'$ is Hermitian with entries in $\mathbb{Z}$.*
(v)   *For any real number $\varphi \in \mathbb{R}$ and any $0 \leq \beta \leq 1$, which can be chosen arbitrarily small, setting*

$$h^{col} := c + \beta c' \qquad \text{independent of } \varphi,$$
$$h^{row}(\varphi) := a + \beta\big(a' + e^{i\pi\varphi}b + e^{-i\pi\varphi}b^\dagger\big),$$

*we have $\|h^{row}(\varphi)\| \leq 2$, $\|h^{col}(\varphi)\| \leq 1$.*

*Define $H^{\Lambda(L)}$ as in Eq. (1), and let $O_{A/B} := L^{-2}\sum_{i \in \Lambda} m_i$. Then, given $\varphi \in [2^{-\eta}, 2^{-\eta} + 2^{-\eta-\ell})$ with $\eta \in \mathbb{N}$, the following statements hold:*

- *If TM halts on input $\eta$, then for some $\ell \geq 1$, $H^\Lambda(\varphi)$ is gapless in the sense of Supplementary Definition 1.2, with a ground state that is critical (i.e. with algebraic decay of correlations), and*

for all eigenstates $|\Psi_B\rangle$ with energy $\langle\Psi_B|H^\Lambda(\varphi)|\Psi_B\rangle \leq 1$ it holds that $\langle\Psi_B|O_{A/B}|\Psi_B\rangle = 0$.

- If TM is non-halting on input $\eta$ and $\ell = 1$, then $H^\Lambda(\varphi)$ is gapped in the sense of Supplementary Definition 1.1, with a unique, product ground state $|\Psi_A\rangle$ with $\langle\Psi_A|O_{A/B}|\Psi_A\rangle = 1$.

The undecidability of which of the two cases pertains follows immediately from the undecidability of the Halting Problem, by choosing TM to be a universal TM. For simplicity, we will refer to the phases A and B determined by the value for the macroscopic observable $O_{A/B}$ as the gapped and gapless phase, respectively.

As a consequence of the new Hamiltonian construction in this paper, we also obtain the following result:

**Corollary 2.2** For all $\varphi \in [0, 1]$, $H^\Lambda(\varphi)$ is either in a phase with a product ground state and a spectral gap $\geq 1$, or it is in a gapless phase with algebraic decay of correlations, where the two phases are distinguished by the expectation value of a macroscopic observable $O_{A/B}$. Moreover, there exists a subset $S \subset [0, 1]$ with Borel measure $\mu(S) > 0$, such that even for computable $\varphi \in S$, determining the phase that $H^\Lambda(\varphi)$ is in is uncomputable.

A less precise but simple interpretation of the above corollary is:

**Corollary 2.3** (informal) The phase diagram of $H^\Lambda(\varphi)$ as a function of its parameter $\varphi$ is uncomputable.

A set of schematic phase diagrams is shown in Fig. 1.

**Constructing the Hamiltonian.** Using well-known methods from the field of Hamiltonian complexity, it is possible to construct a quantum many-body system whose lowest-energy eigenstate represents the evolution of any desired computation[12]. If we introduce a local term in the Hamiltonian that gives additional energy to any state with overlap with the halting state of the computation, we can arrange for states representing computations that halt to pick up additional energy relative to states representing computations that do not halt and open up a gap in the spectrum. In this way, Turing's well-known HALTING PROBLEM can be transcribed into a property of the quantum many-body system, namely whether or not it has a spectral gap. Thus, determining whether the system has a spectral gap is at least as hard as the HALTING PROBLEM. Since the HALTING Problem is known to be undecidable, determining whether the Hamiltonian is gapped or gapless is also undecidable. Conceptually, this is how Cubitt, Perez-Garcia & Wolf, and Bausch et al.[9–11] proved the undecidability of the spectral gap.

The starting point for our construction is also the undecidability of the HALTING PROBLEM[13]: in brief, this states that determining whether a universal (classical) TM (UTM) halts or not on a given input is, in general, undecidable. In the quantum computation setting, Cubitt, Perez-Garcia and Wolf[9] showed how an input can be extracted from a phase in a quantum gate such as $U = \text{diag}(1, \exp(2\pi i\varphi))$, using quantum phase estimation (QPE, ref. [14]), which outputs a binary expansion of $\varphi$. The latter can then be fed as input to a UTM. Thus, this combination of QPE and UTM runs the UTM on any desired input encoded in $\varphi$, and the HALTING PROBLEM for this combination is undecidable.

How do we reduce this QTM-based HALTING PROBLEM to a result about phases in a many-body system? This is a culmination of the following techniques from previous works. However, for each one of them, significant obstacles must be overcome to prove the uncomputability of phase diagrams .

1. The first necessary ingredient is a QTM-to-Hamiltonian mapping, which allows the construction of local, translationally invariant couplings that result in a 1D spin chain Hamiltonian whose ground state energy is exactly zero if the encoded QTM does halts within a certain time interval; or otherwise is positive[15]. Using such an QTM-to-Hamiltonian mapping, a QTM running the QPE + UTM computation described above is encoded into the spin chain Hamiltonian, with $\varphi$ now appearing as a parameter of the resulting Hamiltonian. However, the energy difference between the halting and non-halting cases decreases as the time interval increases, meaning we need further techniques to obtain a non-zero energy difference in the thermodynamic limit.

2. A second ingredient is amplifying this penalty. In ref. [9] this is done by combining the QTM-to-Hamiltonian mapping with an aperiodic tiling Hamiltonian, thereby ensuring that, for each length of a computation, a fixed density of such circuit-to-Hamiltonian instances are distributed across the spin lattice. In this way, the ground state energy density is zero iff the QTM-to-Hamiltonian mapping always has zero energy, and thus depends on whether the QPE + UTM computation ever halts.

3. In ref. [11], point 2 is replaced by a so-called Marker Hamiltonian. This in combination with a circuit-to-Hamiltonian construction results in a ground state, which partitions the spin chain into segments just large enough for the UTM to halt, if it halts. Here, the segments do not have a fixed length, but instead self-adjust to find their own

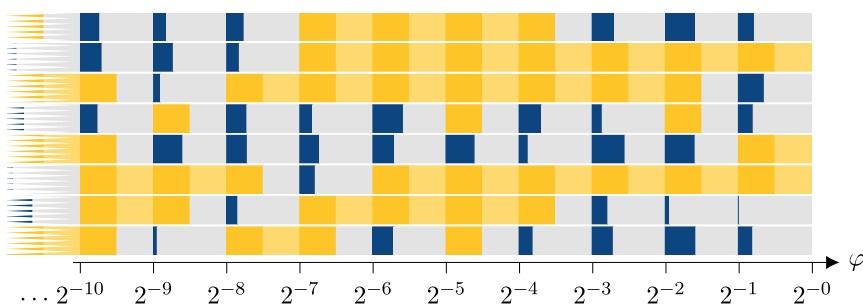

**Fig. 1 A selection of sample phase diagrams of the continuous family $\{H^\Lambda(L)(\varphi)\}_{L,\varphi}$ written for a series of possible universal encoded Turing machines varying from top to bottom, plotted against $\varphi$ on the x-axis (note the log scaling).** Blue means gapless (which is where the TM halts asymptotically on input $\varphi$) and yellow gapped (TM runs forever). At the points $2^{-\eta}$ for $\eta \in \mathbb{N}$, we can have a phase transition between gapped and gapless phases, depending on the behaviour of the encoded TM; there is a positive measure interval above these points where the phase behaviour is consistent. The grey sections are parameter ranges that we do not evaluate explicitly; there will be a phase transition at some point within that region if the bounding intervals have different phases. The lighter yellow area indicates a changing gapped instance. In our construction, the gapless behaviour is more intricately dependent on $\varphi$, but the TM can be chosen such that both halting and non-halting phases cover an order one area of the phase diagram.

length. In contrast to ref. [9], this has the effect that either all encoded instances of computations halt, or none do.

4. The final step is the addition of an Ising-type coupling as in refs. [9,16], which breaks the local Hilbert space up into subspaces $\mathcal{H}_A \oplus \mathcal{H}_B$, and which ensures that the low-energy spectrum is contained either entirely in the A or B subspace, depending on the ground state energy density just constructed. Since determining the ground state energy density is uncomputable, it is also uncomputable to determine whether the system is in phase A or B with respect to the Hamiltonian parameter $\varphi$.

As mentioned, we require significant alterations to this collection of ingredients. Concretely, the issue is that if we encode an input $\varphi$ to be extracted using error-free QPE, then we require the circuit gates to depend explicitly on the binary length of $\varphi$, denoted $|\varphi|$. Consequently, the resulting matrix elements of the Hamiltonian will also explicitly depend on $|\varphi|$—a discontinuous function of $\varphi$. To remove this dependence, we instead perform the QPE procedure approximately, by using a universal gate set that approximates all the gates depending on $|\varphi|$[17]. However, the constructions of Cubitt, Perez-Garcia & Wolf, and Bausch et al.[9,11] crucially rely on the QPE expansion of $\varphi$ to be performed exactly; any errors destroy the constructions.

To overcome this obstacle, we first encode this approximate QPE plus the evolution of a UTM in a QTM-to-Hamiltonian mapping, which has positive energy iff the QPE + UTM computation does not halt. We label the resulting Hamiltonian $H_{comp}$. This is outlined in sections 'Encoding computation in Hamiltonians', 'The encoded computation' and 'From QTM to Hamiltonian' where the QTM-to-Hamiltonian mapping and the computation it encodes are explained, respectively. A significant novel technical contribution of this work is then a proof that the Marker Hamiltonian used in ref. [11] does, in fact, allow for some leeway in the precision to which QPE is performed and can be used to provide a correction for the energy penalty picked up as a result of any errors in the QPE.

To generate the required energy correction, we consider a 2D spin lattice and construct an underlying classical Hamiltonian, which we denote $H_{cb}$, that partitions the lattice into a uniform grid of squares. We note that the method from ref. [9] would be inappropriate for this construction as it would lead to an accumulation of energies we cannot correct for without matrix elements depending explicitly on $|\varphi|$. Within each square, the ground state encodes the evolution of a classical TM (encoded as a tiling problem akin to the ones used in refs. [18,19]), which will calculate the energy correction necessary to offset the error introduced by approximately performing QPE. The classical Hamiltonian is then coupled to the Marker Hamiltonian. We denote this combination $H^{(\boxplus)}$.

Section 'Classical tiling with quantum overlay' describes the ground state of the resulting Hamiltonian $H^{(\boxplus)}$ such that the halting or non-halting behaviour together with the Marker Hamiltonian determines whether the energy density of the constructed Hamiltonian is non-negative (in the non-halting case) or negative (in the halting case). Crucially, it is now robust with respect to the errors present in the expansion of $\varphi$ from the approximate QPE procedure. Finally, in section 'Uncomputability of the Phase Diagram', we show how $H_{comp}$, $H_{cb}$ and $H^{(\boxplus)}$ are combined mathematically to lift this undecidability of the ground state energy density, to uncomputability of the phase diagram, using now-standard techniques[9,11,16]. For a mathematically rigorous derivation, we refer the reader to the Supplementary information.

**Encoding computation in Hamiltonians**. A QTM can be thought of as a classical TM, but where the TM head and tape

configuration can be in a superposition of states. The updates to the QTM and tape configuration are then described by a transition unitary, $U$, describing the transitions of the QTM state and tape, such that the state is updated via the map $|\psi\rangle \mapsto U|\psi\rangle$[20].

Given a QTM, one can construct a local Hamiltonian that has a ground state encoding the evolution of the computation[9,15], closely related to the Feynman–Kitaev Hamiltonian encoding quantum circuits into Hamiltonians[12,21]. The ground state encodes $T$ steps of the computation, where $T$ is a predefined and fixed function of the Hamiltonian's size determined by the particular QTM-to-Hamiltonian mapping. The ground state of such a Hamiltonian is called a *history state* and takes the form

$$|\Psi\rangle = \frac{1}{\sqrt{T}} \sum_{t=1}^{T} |t\rangle |\psi_t\rangle, \qquad (3)$$

where the state of the quantum TM at time step $t$ is $|\psi_t\rangle$. The ground state energy of the Hamiltonian can be made dependent on aspects of the computation by adding a projector that penalises certain computational states, and the resulting energy is known to high precision[22,23].

**The encoded computation**. As in refs. [9,11], the computation we wish to encode via such a QTM-to-Hamiltonian mapping will be a pair of QTMs running in succession: the first will run QPE on a quantum gate $U_\varphi$, which outputs a number in binary, and the second will be a UTM, which takes the output of the QPE as input. The gate $U_\varphi$ is encoded in the transition unitary $U$ describing the QTM, which is in turn encoded in the matrix elements of the Hamiltonian. The energy of the Hamiltonian encoding the computation will then be made dependent on whether the computation halts or not, allowing us to relate its ground state energy to the halting property.

*Phase estimation*. Given a unitary matrix $U_\varphi = \left( 100 e^{i\pi\varphi} \right)$, the QPE algorithm takes as input the eigenvector corresponding to the eigenvalue $e^{i\pi\varphi}$, and outputs an estimate of $\varphi$ in binary. If the number of qubits on which the phase estimation is performed is smaller than the number of bits required to express $\varphi$ in full, the algorithm is only approximate[14]. Furthermore, if a finite gate set is used, some of the required unitary gates in the algorithm must be approximated rather than performed exactly[17]. Hence, from phase estimation, we get an output state consisting of a superposition over binary strings:

$$|\chi(\varphi)\rangle = \sum_{x \in \{0,1\}^n} \beta_x |x\rangle, \qquad (4)$$

where the amplitudes $\beta_x$ are concentrated around those values for which $x \approx \varphi$ and rapidly drop off away from $\varphi$. Details are in Supplementary Note 2.

*Universal QTM*. We then feed the output $|\chi(\varphi)\rangle$ of this phase estimation into the input of a universal TM, as in ref. [9], which then runs a computation that may or may not halt. By the well-known undecidability of the Halting Problem[13], determining whether the QTM halts for a given string is undecidable.

**From QTM to Hamiltonian**. Using the QTM-to-Hamiltonian mapping described in section 'Encoding computation in Hamiltonians', the computation outlined above is mapped to a 1D, translationally invariant, nearest-neighbour Hamiltonian $H_{comp}(\varphi)$[15], with a penalty for the non-halting case. It can be shown that the ground state energy of $H_{comp}(\varphi)$ scales as

$$\lambda_{min}(H_{comp}(\varphi)) \sim \epsilon(L)/\text{poly } L, \qquad (5)$$

where

$$\epsilon(L) = \sum_{x \in S(L)} |\beta_x|^2. \qquad (6)$$

The $\beta_x$ are the QPE coefficients in Eq. (4), and $S(L)$ is the set of inputs for which the universal TM does not halt within time $T(L)$.

Since the $\beta_x$ are concentrated around the binary expansion of $\varphi$, if the latter encodes a halting instance, there will be a length $L_0$ for which $\epsilon(L) \approx 0$ for all $L > L_0$; otherwise, $\epsilon(L) \approx 1$ for all $L$. This immediately yields a Hamiltonian for which the ground state energy is halting-dependent (and hence uncomputable). We refer the reader to Supplementary Note 3 for details.

**Tiling and classical computation**. In Eq. (5), we see that the difference between the Hamiltonian's ground state energy in the case where $\epsilon(L)$ from Eq. (6) is ~1 or 0 decreases with the system size $L$. Thus, the energy gap between the two cases goes to zero irrespective of whether $\varphi$ encodes a halting or non-halting instance. To amplify this gap so that there is a finite energy gap in the thermodynamic limit (as per points 2 and 3), we will combine the Feynman–Kitaev Hamiltonian with a classical Hamiltonian based on a Wang tiling that partitions the space suitably to ensure a fixed density of computation instances is spawned across the lattice. The result we achieve with this is an energy gap opening up as $L$ grows between the cases where $\epsilon$ takes different values.

A set of Wang tiles—square, 2D tiles with coloured sides, with the rule that adjacent sides of neighbouring tiles in a tiling of the plane must have matching colours—can be mapped to a classical Hamiltonian: if tiles $t_i$, $t_j$ cannot be placed next to each other, then we introduce a term $|t_i\rangle\langle t_j|$ into the Hamiltonian. The overall Hamiltonian is the sum over all such local terms, such that its ground state represents a tiling satisfying the tiling rules (if such a tiling exists). If no such tiling exists, the ground state energy is ≥1.

Similarly, it is well known that there exist tile sets that encode the evolution of a classical TM[18,19] within a square grid: TM tape configurations are represented by rows, such that adjacent rows represent successive time steps of the TM (Fig. 2).

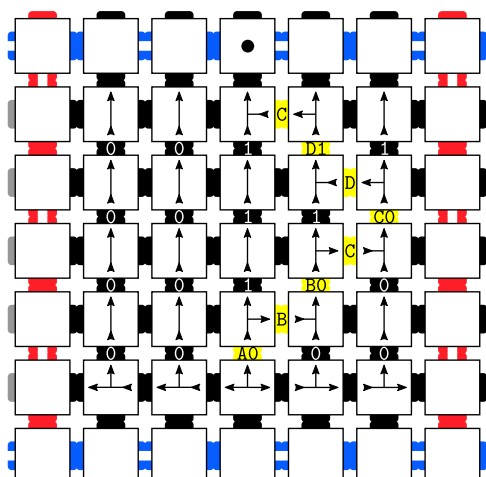

**Fig. 2 The evolution of a classical TM can be represented by Wang tiles, where colours of adjacent tiles have to match, and arrowheads have to meet arrow tails of the appropriate kind.** Here, the evolution runs from the bottom of the square to the top, where it places a marker ● on the boundary as explained in section 'Classical tiling with quantum overlay'. In this image, the TM's evolution is contained in an individual square in the checkerboard grid shown in the figure below.

We combine both Wang tiles and the TM tiling ideas by constructing a tile set whose valid tilings have the following properties:

1. A tiling pattern that creates a square grid across the lattice Λ (much like a checkerboard). The side length of the grid squares can be varied (provided all grid squares are the same size) and still correspond to a valid tiling (depicted in Fig. 3).
2. Within each square of the grid, we use the tiles to encode a TM that first counts the size of the square it is contained in, and then outputs a marker ● on the top border of the square, where the placement of this marker is a function of the size of the square (depicted in Fig. 2).

Having developed this tiling, we map it to a corresponding tiling Hamiltonian, which we denote $H_{cb}$, such that its ground states retain the properties of the valid tilings listed above. The reader is referred to Supplementary Note 4 for details.

**Classical tiling with a quantum overlay**. We now want to combine the classical Hamiltonian encoding the Wang tiles, and the quantum Hamiltonian encoding the HALTING PROBLEM computation, to create an overall Hamiltonian that has a large ground state energy difference between the halting and non-halting cases, without the ~1/poly($L$) decay in Eq. (5). To do so, we split the local Hilbert space of each lattice spin into a classical part $\mathcal{H}_c$ and a quantum part $\mathcal{H}_e \oplus \mathcal{H}_q$ giving $\mathcal{H} = \mathcal{H}_c \otimes (\mathcal{H}_e \oplus \mathcal{H}_q)$, where $\mathcal{H}_e = \{|e\rangle\}$ just contains a filler state $|e\rangle_e$. The ground state can then be designed to be a product state of the form $|C\rangle_c \otimes |\psi_0\rangle_{eq}$, where $|C\rangle_c$ is a valid classical tiling configuration—as described in section 'Tiling and classical computation'—and $|\psi_0\rangle_{eq}$ is a quantum state with the following properties:

1. We use the 1D Marker Hamiltonian from ref. [11], and couple its negative energy contribution to the size of each grid square in the classical tiling and the placement of the ● marker. The negative energy each square contributes is a determined by where the ● marker is placed, and thus by the action of the classical TM. We denote this combined Hamiltonian $H^{(\boxplus)}$.
2. We effectively place the ground state of a Hamiltonian $H_{comp}$ encoding the QPE plus universal TM along the top edge of the square, by adding additional penalty terms to the Hamiltonian that penalise the classical and quantum layers to occur in this configuration elsewhere.
3. Everywhere not along the horizontal edge of a grid square in $\mathcal{H}_c$ is in the zero-energy $|e\rangle_e$ filler state in $\mathcal{H}_e \oplus \mathcal{H}_q$.

As mentioned, the patterns in the degenerate ground space of $H_{cb}$ are checkerboard grids of squares with periodicity $w \times w$, where the integer square size $w$ is not fixed.

By choosing the classical TM encoded in the tiling to place a ● marker at an appropriate point, we are able to tune the ground state energy of $H^{(\boxplus)}$ such that the total energy of a single $w \times w$ square $A$ in the checkerboard pattern is:

$$\lambda_{min}(w) := \lambda_{min}\left(H^{(\boxplus)}\Big|_A + H_{comp}\Big|_A\right)$$
$$\begin{cases} \geq 0 & \text{if } \epsilon(w) \geq \epsilon_0(w) \, \forall w, \\ < 0 & \text{if } \epsilon(w) < \epsilon_0(w) \, \forall w \geq w_0, \end{cases} \qquad (7)$$

where $\epsilon_0(w)$ is some cut-off point, $w_0$ is the halting length (recall from the previous section that the runtime of the computation encoded in the ground state depends on the size of the available tape, i.e. the size of the checkerboard square edge that the TM runs on), and where $\lambda_{min}(w_0) = -\delta(w_0) < 0$ for the halting length $w_0$ is a small negative constant.

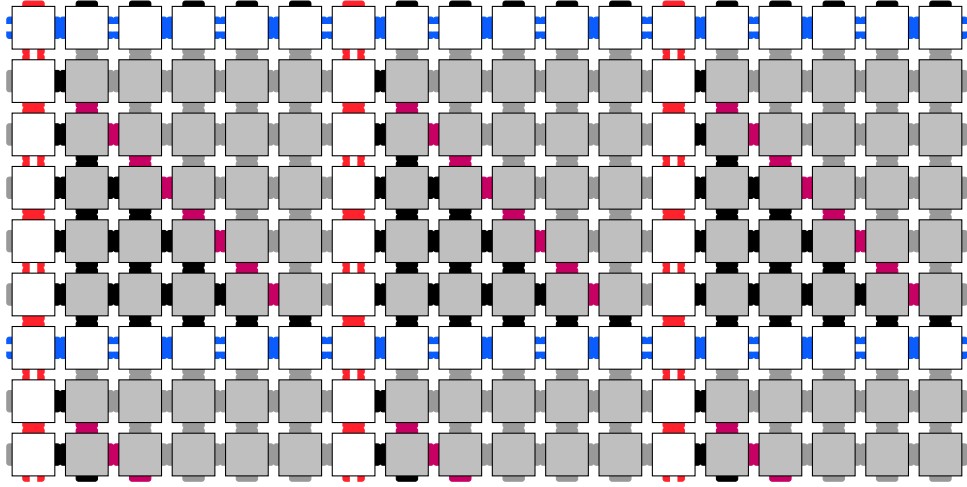

**Fig. 3 Section of the checkerboard tiling Hamiltonian's ground state.** The white squares form borders, and in the interior, we place tiles simulating the evolution of a classical Turing Machine.

The Marker Hamiltonian's energy bonus thus compensates for the QPE approximation errors by lowering the energy by just enough such that a halting instance has negative energy. On the other hand, the energy of the non-halting instance remains large enough that the energy of a single square remains positive[11,23].

Thus, provided $\epsilon(w)$ is sufficiently small, the ground state of $H_{cb} + H_{comp} + H^{(\boxplus)}$(+coupling terms) is a checkerboard grid of squares with a constant but negative energy density. Otherwise, the ground state energy density of the lattice is lower bounded by zero. Which of the two cases holds depends on determining whether $\epsilon(w) \geq \epsilon_0(w)$ or $< \epsilon_0(w)$, which is undecidable; undecidability of the ground state energy density follows. The reader is referred to Supplementary Note 5 for more details.

We define the Hamiltonian formed by $H_{cb}$, $H_{comp}$ and $H^{(\boxplus)}$ and the coupling terms as $H_u(\varphi)$. Assume $\varphi$ encodes a halting instance and set $w = \text{argmin}_s\{\lambda_{\min}(s) < 0\}$, and A is a single square of size $w \times w$. Then, the ground state energy $H_u(\varphi)$ on a grid $\Lambda$ of size $L \times H$ is given by

$$\lambda_{\min}(H_u(\varphi)) = \left\lfloor \frac{L}{w} \right\rfloor \left\lfloor \frac{H}{w} \right\rfloor \lambda_{\min}(H_u(\varphi)|_A). \tag{8}$$

**Uncomputability of the phase diagram.** To go from undecidability of the ground state energy density, demonstrated at the end of section 'Classical tiling with quantum overlay' to the undecidability of the phase (and spectral gap), we follow the approach of Cubitt, Perez-Garcia and Wolf[9] by combining $H_u(\varphi)$ with a trivial state $|0\rangle$ such that $|0\rangle^{\otimes\Lambda}$ has zero energy, and the spectrum of $H_u(\varphi)$ is shifted up by $+1$ (Supplementary Lemma 6.5). From this shift and Eq. (8) it can be shown that

$$\lambda_{\min}(H_u(\varphi)) \begin{cases} \geq 1 & \text{in the non-halting case, and} \\ \longrightarrow -\infty & \text{otherwise.} \end{cases} \tag{9}$$

Let $h_u(\varphi)$ denote the local terms of $H_u(\varphi)$, let $h_d$ be the local terms of the critical XY model, and let $|0\rangle$ be a zero energy state, such that the total Hilbert space is $(\mathcal{H}_1 \otimes \mathcal{H}_2) \oplus \{|0\rangle\}$. Then, the local terms of the total Hamiltonian $H^\Lambda(\varphi)$ are defined as

$$h_{i,i+1}(\varphi) := |0\rangle\langle0|^{(i)} \otimes (\mathbb{1} - |0\rangle\langle0|)^{(i+1)}$$
$$+ (\mathbb{1} - |0\rangle\langle0|)^{(i)} \otimes |0\rangle\langle0|^{(i+1)} \tag{10}$$

$$+ h_u^{i,i+1}(\varphi) \otimes \mathbb{1}_2^{(i)} \otimes \mathbb{1}_2^{(i+1)} + \mathbb{1}_1^{(i)} \otimes \mathbb{1}_1^{(i+1)} \otimes h_d^{i,i+1}. \tag{11}$$

The result is the following: the overall spectrum of the Hamiltonian is

$$\text{spec}\left(H^\Lambda(\varphi)\right) = \{0\} \cup \left(\text{spec}\left(H_u(\varphi)\right) + \text{spec}\left(H_d\right)\right) \cup G,$$

where $G \subset [1, \infty)$. To understand this, we consider the two cases. If a non-halting instance is encoded $\lambda_{\min} \geq 0$ in Eq. (7), then $H_u(\varphi)$—from Eq. (9)—has ground state energy lower-bounded by 1; the ground state of the overall Hamiltonian is the trivial zero energy classical product state $|0\rangle^{\otimes\Lambda}$, and $H^\Lambda$ has a constant spectral gap. If $\lambda_{\min} < -|\delta|$, then $H_u(\varphi)$ has a ground state with energy diverging to $-\infty$. We further note that the critical XY model has a dense spectrum with zero ground state energy, hence from the $H_d$ term we obtain a dense spectrum above the ground state[24]. As a result, the Hamiltonian becomes gapless and has a highly entangled ground state with algebraically decaying correlations.

Since the existence of a halting length $w_0$ in Eq. (7) is undecidable, discriminating between $\lambda_{\min} \geq 0$ or $< -|\delta|$ is also undecidable. This implies determining whether the Hamiltonian is in the critical, quantum-correlated phase or the trivial product state gapped phase is undecidable as well.

As $H^\Lambda(\varphi)$ is a continuous function of $\varphi$, there exist finite measure regions for which all values of $\varphi$ have the same ground state and for which there is no closing of the spectral gap, which delineates the two phases. Setting $\Pi_i := |0\rangle\langle0|^{(i)}$, the observable $O_{A/B} = L^{-2}\sum_{i \in \Lambda}\Pi_i$ has expectation value 1 when in the state $|0\rangle^{\otimes\Lambda(L)}$, and 0 in the other case. This is true even if the observable is restricted to a finite geometrically local subset of the lattice. We refer the reader to Supplementary Note 6 for more details.

## Discussion

Our result proves the undecidability of the phase and spectral gap for a continuous, one-parameter family of Hamiltonians on a two-dimensional lattice. An immediate consequence is that there is no algorithm that can compute a Hamiltonian's phase diagram in general, even given a complete description of its microscopic interactions.

Qualitatively, this brings the results close to classic condensed matter models, for example, the transverse Ising model described by the Hamiltonian $H_{TIM} = \sum_{\langle i,j\rangle}\sigma_z^{(i)}\sigma_z^{(j)} + \varphi\sum_i\sigma_x^{(i)}$. Here, the real parameter $\varphi$ determines the strength of an external magnetic field. Its phase diagram comprises an ordered and a disordered

phase, with an order parameter given by the macroscopic observable $O_{\mathrm{TIM}} = \frac{1}{N} \sum_{j=1}^{N} \sigma_z^{(j)}$ corresponding to the global magnetisation per spin. In the ordered phase, the ground state expectation value $|\langle O_{\mathrm{TIM}} \rangle| = 1$. In the disordered phase, the ground state expectation value is 0. Both these phases have a non-zero spectral gap in the thermodynamic limit. At the critical point $\varphi = 1$ between the two phases, the system is gapless and exhibits criticality.

As mentioned previously, we can take as the order parameter the macroscopic observable $O_{\mathrm{A/B}} = \frac{1}{N} \sum_j \Pi_{\mathrm{A}}^{(j)}$, where $\Pi_{\mathrm{A}}^{(j)}$ is a projector onto the local Hilbert space $\mathcal{H}_{\mathrm{A}}$ on lattice site $j$. This definition yields a familiar picture: for those $\varphi$ such that $H(\varphi)$ is in the A phase, we prove that the ground space is non-degenerate with $\langle O_{\mathrm{A/B}} \rangle = 1$. In the B phase, all ground states have an expectation value $\langle O_{\mathrm{A/B}} \rangle = 0$ with respect to this observable. Thus, phases A and B are distinguished by an order parameter given by a macroscopic observable $O_{\mathrm{A/B}}$, and the system undergoes a (first-order) phase transition between these as $\varphi$ varies.

However, unlike the Ising model, in our case, one phase (phase A, say) is gapped, but the other (phase B) is a gapless phase. This gapped vs. gapless phase transition has further phenomenological consequences. For instance, a transition between A and B implies a transition from exponential decay of correlations to long-range correlations with algebraic decay of correlation functions. In fact, something stronger holds in our case: in the gapped phase, the ground state is a product state and all connected correlation functions are strictly zero.

If we define a phase diagram of a $k$-parameter Hamiltonian as a normalised parameter space $[0, 1]^k$, which maps out the different phases as a function of the parameters at each point $p \in [0, 1]^k$, then previous undecidability results[9,11] do not imply uncomputablity of phase diagrams, for multiple reasons. There, as here, the matrix elements of the local interactions of the Hamiltonian $H(\varphi)$ depend on an external parameter $\varphi$, which determines the gappedness of the Hamiltonian. However, importantly, in the previous constructions, the matrix elements also depend on the binary length of $\varphi$, denoted $|\varphi|$, which is a discontinuous function of $\varphi$. A consequence of this, it is not possible to define a meaningful phase diagram for these Hamiltonians over the required parameter range. This significantly limits the implications one can draw from previous spectral gap undecidability results, in particular for quantum phase diagrams, which are one of the main reasons for caring about spectral gaps in the first place.

Although the construction developed herein proves undecidability between phases defined by $O_{\mathrm{A/B}}$, the result can be extended to more general phase diagrams by a small modification to the construction. If we modify the Hamiltonian by introducing two terms $h_{\neg X}$ and $h_X$, which are Hamiltonians that, respectively, have and do not have the ground state property $X$ in the thermodynamic limit (specifically, in Eq. (10) defining the Hamiltonian, we replace $|0\rangle\langle 0|$ and $h_d$ with $h_{\neg X}$ and $h_X$), then the new overall Hamiltonian will have two phases, one of which has property $X$ and another which does not. Determining which of the two properties holds is undecidable.

Furthermore, the algorithmic uncomputability of the phase diagram problem implies axiomatic independence of the problem[25]. That is, for any consistent formal system with a recursive set of axioms, there exists a Hamiltonian of the form given in Theorem 2.1 such that determining the phase diagram from the given axioms is not possible.

There are other consequences: a common technique in numerical condensed matter physics to estimate the phase of a physical system is to take the Hamiltonian on an $L \times L$ lattice, calculate the phase for this lattice size by some numerical means, and then extrapolate its phase to the thermodynamic limit. This is justified by the assumption that as long as $L$ is sufficiently large, the system already displays the behaviour of the thermodynamic limit. In ref. [10] it was shown that this assumption is not justified in all cases, as the phase may without warning appear completely different at some arbitrarily large and uncomputably large system size. Leading to the phenomenon of sized-driven phase transitions explored in ref. [16]. Our result further extends this to show that attempting to compute the phase diagram by extrapolating from some finite-size system may not reflect the phase diagram in the thermodynamic limit.

We note, however, that our result only establishes uncomputability for certain highly complex and artificially constructed Hamiltonians. Furthermore, the Hamiltonians constructed by our techniques are necessarily frustrated. For many commonly occurring Hamiltonians—particularly those with small local Hilbert space dimension—determining the phase may well be rigorously decidable. For example, using techniques from refs. [26,27], as done for example in ref. [28], completely solves the case of frustration-free, nearest-neighbour 1D qubit chains.

As aforementioned, previous gap undecidability results[9,11] required the explicit inclusion of the binary length of $\varphi$, that is, $|\varphi|$, as matrix elements of the form

$$2^{-2|\varphi|} \qquad \text{or} \qquad e^{-i\pi 2^{-2|\varphi|}}.$$

It is clear that one cannot vary $\varphi$ along a continuous path between two points $\varphi_1$ and $\varphi_2$ while keeping the length of its binary expansion $|\varphi|$ fixed at all points along the path. Moreover, varying $\varphi$ and $|\varphi|$ separately breaks the construction. As a result, it is impossible to draw a phase diagram with respect to the parameter $\varphi$ for the Hamiltonians of refs. [9,11].

Phase transitions are typically defined as points at which there is a non-analyticity in the ground state energy (or some associated order parameter) with respect to continuous changes in $\varphi$. If one were to view $|\varphi|$ as an explicitly discontinuous function of $\varphi$, the ground state may no longer analytically depend on $\varphi$ even at points where the system is gapped. As such it is unclear if it is even meaningful to define a phase or phase transition for the models presented in refs. [9,11]. In addition, Hamiltonians depending discontinuously on a continuously varying parameter are not typically encountered in physics.

The family of Hamiltonians we construct in this work is truly continuous, that is, we define our local terms $h_{\mathrm{row}}(\varphi)$ for arbitrary $\varphi \in \mathbb{R}$, thus even irrational numbers with infinitely long binary expansions are perfectly fine as instances of our problem set-up. This brings us qualitatively closer to models of Hamiltonians of real systems, where the parameter varied will typically be some physical property, such as an applied magnetic field, which can be varied continuously.

Since Theorem 2.1 shows that for any $\varphi$ there exists a small finite interval around $\varphi$ for which the phase of the Hamiltonian is the same, we have a notion of stability of undecidability under perturbations to $\varphi$—something that was not the case in previous results. However, it is not clear if there is any stability of the Hamiltonian's properties with respect to perturbations in arbitrary matrix elements. Stability of undecidability under arbitrary local perturbations to the Hamiltonian remains a challenging but important topic for future research, but one that has yet to be fully resolved even for simple models such as the Ising model. Finally, it important to emphasise that the Hamiltonian constructed here is highly artificial, in the sense that it has an unnaturally large local Hilbert space dimension and highly complex, specifically tailored interactions. While size-driven phase transitions have been discovered in much simpler models[16], and recent results in Hamiltonian complexity theory[29–32] show that related complexity-theoretic properties can also occur

in far simpler models, it remains an open question whether undecidability occurs in any remotely natural Hamiltonians.

Ongoing research directions both in Hamiltonian complexity and computability focus on reducing the physical dimensionality of the system, reducing the local Hilbert space dimension and choosing physical interactions comparable to those seen in physical systems. A further route of investigation would be to determine how difficult it is to compute phases for systems of finite size, rather than in the thermodynamic limit, for some suitable definition of phase in the finite-size setting.

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

## Acknowledgements

J.B. is supported by the Draper's Research Fellowship at Pembroke College. J.D.W. is supported by the EPSRC Centre for Doctoral Training in Delivering Quantum Technologies [EP/L015242/1]. T.S.C. is supported by the Royal Society. This work was supported by the EPSRC Prosperity Partnership in Quantum Software for Simulation and Modelling (EP/S005021/1).

## Author contributions

J.D.W., J.B. and T.S.C. all contributed significantly to the research.

## Competing interests

The authors declare no competing interests.
