## [Peer Review File · Nature Communications]

REVIEWER COMMENTS

Reviewer #1 (Remarks to the Author):

Summary:

This paper studies the computability of the phase diagram of a quantum material at zero-temperature in the thermodynamic limit. There is a folklore heuristic for determining the phase of a material by brute-force simulating it for large but computationally feasible sizes and then concluding the phase remain as predicted for all larger systems, and in particular in the thermodynamic limit. This paper provides a provable example of an explicit material for which the mentioned heuristic fails. They design a two-dimensional translationally invariant Hamiltonian depending on a single real valued parameter for which determining whether the system has a constant level of spectral gap is not decidable by Turing machines. In their construction, the gapless regime corresponds to a critical phase with significant quantum correlations across the ground state. Whereas, the gapped phase has a classical description.

On a technical level, this work mainly builds off two previous works of Cubbitt-Peres-Garcia-Wolf (CPGW) and Bausch-Cubitt-Lucia-Perez-Garcia (BCLPG). The former establishes undecidability of the spectral gap for a discrete two-dimensional family of Hamiltonians. The latter improves the former to a discrete family of one-dimensional Hamiltonians. The main contribution of this work is to establish the same level of undecidability for two-dimensional but continuous family of Hamiltonians. In particular, they show undecidability holds for a positive measure of Hamiltonians.

The introduction Section is nicely presented. Although, the designed model is further from natural, the claim of this result, however, is a great progress towards the goal of establishing undecidability for more natural families.

Evaluation:

I believe the topic is an interesting one and the claimed result is an important one and would benefit a wide range of audience. However, the write-up in its current form is not very accessible and needs to be improved significantly before I can hold a judgement regarding publication.

I try to explain a bit more about my point on the write-up. Firstly, a high-level and intuitive explanation of the Hamiltonian construction is missing. Section 3 attempts to provide such interpretation. However, it is hard to follow, in my opinion. And it does not quite give a background on how the previous constructions of CPGW and BCLPG work, what are their limitations in showing the current result, how you overcome those challenges, and what new tools they develop relative to previous work etc.

Second, more organization across and within the sections would make the content more accessible. For instance, in several places the goal of each section is not clearly stated at the beginning of each section, and a reader may feel lost figuring out where the content is headed. Section 3 in particular needs to be improved by adding an overall picture and organization to it. Currently, it is a collection of disjoint topics without a coherent theme gluing them together. The supplementary material also has very little organization which makes it difficult to verify the result; adding cross reference between main and supplementary sections may help in grasping the proof.

Thirdly, since this paper is based on several previous developments, many essential tools/concepts are borrowed from those previous works. In my opinion, more attempt is needed to explain what these tools are at least on a high-level. As a reader, I expect to follow the details, at least on a conceptual level, by reading the paper by itself and consult the previous works for further details. My impression is that the paper may not be accessible to an audience who has not studied to the previous CPGW and BCLPG constructions closely. Lastly, there are several vague sentences and simple errors across the text.

Comments:

Some of these points get resolved as one keeps reading the paper. However, on the first round of reading, a reader may find the following points confusing.

1. Line 278: It is impossible to encoding → It is impossible to encode
2. Line 118: 'axiomatic and algorithmic sense' too vague.
3. Different pieces in Section 3.2 seem disjoint. It would be easier for the reader to follow if you first give a coherent picture about how these pieces are going to be connected. For example, starting the section it is not clear whether the section is sketching the proof or just presenting disjoint pieces to be put together later.

4. 3.2 Phase estimation: I assume as input you need a specification of the eigenvalue, e.g., corresponding eigenvector etc.
5. In Section 3.2: 'interactions depending on ϕ and not $|\phi|$ ' is vague. Perhaps, you mean something like continuously depends on ϕ ?
6. It would be worth mentioning in 3.4 that the Wang tiling problem is undecidable.
7. Line 263: '.. is, cannot be ..' probably a typo?
8. Line 271: stability in perturbation of ϕ is not mentioned in the main text but highlighted in the discussions section.
9. Hard to follow what the purpose of Section A is. E.g., I would add more introductory remarks before the content presented in A1.
10. Line 415; is definition R_n correct?
11. I don't follow 425 and eq above. Also eqn right below 445.
12. 1233 is not a sentence
13. Line 39 may have a misplaced comma.
14. Line 47: does \rightarrow does not?
15. Is superconductivity an example a phenomenon related to the characterization of the phase of matter discussed in this paper. My impression is that this paper is about zero-temperature phase transitions.
16. Line 26. `;` \rightarrow `:`.
17. Line 47: 'logically impossible' is somewhat vague
18. Line 110: 'there exists a halting instance η ..' is too vague. What does it mean for η being a halting instance of M ? The wording can be clarified to include the construction interprets ϕ as a string of bits representing the description of some TM relative to some universal TM.
19. Before Corollary 26 it would make sense to describe what a phase diagram is in more precise terms.
20. Section 3.1 has no introductory sentences. It is not clear what is going to happen.
21. Line 140: It would help to say a sentence or so clarifying what QTM is. From what it can be perceived, the way QTM is used is as if it is quantum circuit. E.g. what if the QTM does not halt? The state in equation (3) does not make sense.
22. The paragraph after line 158: you explain QPE and at the end you end up with a quantum state $|\chi(\phi)\rangle$. But what U_ϕ is used here? Plus the input of a QPE must include an eigenvector.
23. Section 3.3 is not possible to follow.

24. Line 189: 'We now want to combine classical and quantum Hamiltonians together.' Which classical and quantum Hamiltonians?
25. Line 271: I don't recall any discussion about stability in ϕ before this sentence.
26. Line 132: "the TM". Which TM?
27. Line 735: 'proof' \rightarrow prove

Reviewer #2 (Remarks to the Author):

This manuscript addresses the question of the computability of the phase diagram of a quantum mechanical system defined on a lattice with short-range interactions. Its main results can be found in Theorem 2.4 and its immediate corollaries, and these results answer the question negatively. More precisely, the main theorem states the undecidability of the spectral gap problem for a continuous family of Hamiltonians.

This result must be compared with a similar result [CPGW15a] (in Nature, 2015) and [CPGW15b], with one of the authors common to all publications. The current manuscript makes a very honest statement in this comparison by pointing out that the novelty of the present result is in the possible continuity of the interaction as a function of the parameter ϕ . All three results are constructive, namely the proof proceeds by exhibiting an explicit family of interactions for which the question of the spectral gap is formulated as a halting problem. In the former cases, the interactions were not continuous, while they are in the present manuscript.

This is indeed an interesting result. The novelty of continuity is indeed more than just a technical issue, when seen from the point of view of the phase diagram rather than from that of the spectral gap. However, I am not convinced that Nature Communications is well-suited for the publication of this result. The conceptual and inspiring breakthrough was made in the previous articles mentioned above. The novelty here is of interest to a much more restricted group of quantum information scientists, which makes this manuscript more suited to a more focused journal.

I should also say that I am somewhat confused by the very statement of the main corollary, Corollary 2.5. What exactly is meant by 'the phase of $H(\phi)$ is uncomputable'? In fact, what is 'the phase of $H(\phi)$ '? Since this is in fact the central claim of the paper, it should also be perfectly clear.

Referee report for Nature Communications

Article: “*Undecidability of Phase Diagrams*”

Authors: Johannes Bausch, Toby S. Cubitt, James D. Watson

Contents. The manuscript under review constructs a family of translation-invariant and local quantum spin Hamiltonians $H(\varphi)$ which depend continuously on a parameter $\varphi \in [0, 1]$ such that the question whether $H(\varphi)$ is gapped or gapless is equivalent to the undecidable halting problem for a universal Turing machine on input φ .

This is the latest in a series of recent results [1, 3] which show that the undecidable halting problem can be embedded into the spectral gap problem of quantum many-body theory. As in all of these works, the result is more of in-principle than practical nature: the constructed Hamiltonian (while translation-invariant and local and therefore physical) is highly artificial, especially because of its very large local dimension, and has little in common with concrete models of interest to the wider physics community. Still, these developments have theoretical appeal and have also led to the interesting notion of system-size driven phase transitions [2].

In the present manuscript, a technical limitation of the previous works is overcome. Namely, in the previous works the Hamiltonian depends on the discontinuous function $|\varphi|$, defined as the length of the binary expansion of the parameter $\varphi \in [0, 1]$. Here, this is remedied by analyzing a variant of the quantum phase estimation algorithm which is based on a different, unary expansion of φ (see Definition A.1) and the Solovay-Kitaev algorithm to approximate small rotation gates and therefore the inverse Quantum Fourier Transform that previously necessitated the use of $|\varphi|$. This approximate quantum phase estimation introduces a small error that is then compensated by a clever definition of a 2D marker Hamiltonian drawing on ideas from [1].

Summary. Altogether, the paper overcomes a relevant technical difficulty (namely discontinuity in the underlying parameter) of recent works on related problems. The arguments appear sound and are well-presented. Equally importantly, the main technical contribution has physical content as it establishes the uncomputability of the gapped versus gapless phase diagram of the constructed Hamiltonians; see Corollary 2.5 and the discussion in Section 4 why continuity is essential for this.

While I do have a few complaints about the presentation (see below), the result is of fundamental nature and adds appreciably to the existing literature. Therefore, I recommend this manuscript for publication in Nature Communications provided the following issues are satisfactorily addressed.

Issues to be addressed. My main complaint is that the notion of phase diagram is used too broadly in the title, abstract, and introduction and this can easily lead to misunderstandings about what is really shown here. While the abstract references high-temperature superconductivity (a seemingly random example), the precise meaning of the term “phase diagram” in the present context is only given later, in Corollary 2.5. To be clear, most physicists define a “phase transition” as a discontinuous change of a macroscopic observable (like total magnetization or one of its derivatives) or as a change of a discrete topological index. While high-temperature superconductivity falls into this category, it is less common to define a “quantum phase” directly in terms of the existence or absence of a gap. (Of course, the closing of a bulk spectral gap indicates the possibility of a change in the topological behavior of the ground state, but (a) the topological property is really what defines the nature of the phase, not the gap, and (b) the gaplessness in that scenario occurs at the phase transition point which is not usually thought of as a separate phase.)

This is all to say that the authors prove uncomputability of the “gapped versus gapless phase diagram”. While this is a diagram showing different types of correlation behavior in the system (and thus a phase diagram in a much looser sense of the word), it is not a phase diagram in the other sense described above and the abstract and introduction should make this fact much more clear than they presently do.

A second complaint I have is that the authors should probably not sell the implications of their result on the empirical method of studying larger and larger volumes to extrapolate to thermodynamic behavior as strongly as they do. The constructed Hamiltonian is highly artificial and the proven undecidability result almost certainly has no bearing on the Hamiltonians that are actually studied in this way.

In fact, for frustration-free spin systems, such as AKLT models, there even exist rigorous finite-size criteria for spectral gaps in two-dimensions which depend continuously on underlying parameters and can thus be used to identify gaps for a range of parameter values. While these methods have their limitations, if one comes across a random model in research it is far more likely that these general methods apply and one can infer rigorous information from finite-volume behavior than that the gapped versus gapless phase diagram is uncomputable. I actually think that the authors, all very respectable researchers, agree with me on these points and I would just suggest that they explain them more clearly.

Below is a list of minor corrections and comments for the consideration of the authors.

1. The convention that $|\varphi|$ denotes the length of the binary expansion of φ should be stated when $|\varphi|$ first appears on page 2, whereas right now it is

first stated in Section 4.

2. At the top of page 7, the fundamental and pioneering nature of [4] could be made more clear than it is right now.
3. The footnote on page 10 has a typo: “length of the” is written twice.
4. Even if it is very standard, it should be noted once that QFT stands for Quantum Fourier Transform.
5. It would be of benefit to the wider community if the authors could do a better job at comparing their techniques and ideas to the ones in the previous works [1, 3]. Given the overlap of authors between these works, this is presumably easy to do and I believe it would help others to identify the key methodological differences. Only the section on quantum phase estimation is fully clear in this regard. As a concrete example, it would be helpful to explain how the tiling and marker Hamiltonian used here differ from the implementation via the quasiperiodic Robinson tiling in [3].

References

- [1] J. Bausch, T. S. Cubitt, A. Lucia, and D. Perez-Garcia, Undecidability of the spectral gap in one dimension. arXiv preprint arXiv:1810.01858 (2018)
- [2] J. Bausch, T. S. Cubitt, A. Lucia, D. Perez-Garcia, and M. M. Wolf, Size-Driven Quantum Phase Transitions, *Proc. Natl. Acad. Sci. (USA)* 115 (2018), pp. 19–23
- [3] T. S. Cubitt, D. Perez-Garcia, and M. M. Wolf, Undecidability of the spectral gap, *Nature* 528 (2015), p. 207
- [4] D. Gottesman and S. Irani, The quantum and classical complexity of translationally invariant tiling and Hamiltonian problems, 50th Annual IEEE Symposium on Foundations of Computer Science, FOCS’09. 2009, pp. 95–104.

Authors' Response: Uncomputability of Phase Diagrams

Johannes Bausch^{*1}, Toby S. Cubitt², and James D. Watson^{†2}

¹CQIF, DAMTP, University of Cambridge, UK

²Department of Computer Science, University College London, UK

July 31, 2020

Summary We have made major revisions to the original draft submitted. These changes can be seen explicitly in the file “Manuscript_Diff.pdf” which is included in the resubmission. The main revisions are:

1. We have clarified that the phase of the Hamiltonian can be defined in the standard way in terms of a macroscopic order parameter, rather than gapped vs gaplessness. We make it clear that when going between the two phases there is a discontinuous change in the expectation value of this macroscopic observable (as well as a closing of the spectral gap), and it is this observable which we use to characterise the phase.
2. A preamble has been added to section 3 which gives a brief overview of the overall proof. A subsection 3.1 has been added, which compares techniques from previous results to the ones used in this work, and also gives a high-level overview of the proof and techniques involved.
3. We have tried to clarify the proof overview in section 3 and its subsections by removing some of the mathematics and emphasising how the different sections relate.

*Email: jkrb2@cam.ac.uk

†Email: ucapjdj@ucl.ac.uk

Reviewer 1 Corrections

Broad Comments:

1. *“Firstly, a high-level and intuitive explanation of the Hamiltonian construction is missing. Section 3 attempts to provide such interpretation. However, it is hard to follow, in my opinion. And it does not quite give a background on how the previous constructions of CPGW and BCLPG work, what are their limitations in showing the current result, how you overcome those challenges, and what new tools they develop relative to previous work etc.”* — we have added a paragraph at the beginning of section 3 with a high level overview of how the construction works. This is then followed by a slightly more detailed overview in section 3.1 which also compares our results/techniques with those of two previous works on undecidability in quantum many-body physics. This subsection also explains difference in techniques between this work and previous works, and explains why these differences matter with regards to proving uncomputability of phase diagrams. In section 4 we have also added a more in-depth discussion of how our techniques and results differ from the previous results.
2. *“Second, more organization across and within the sections would make the content more accessible. For instance, in several places the goal of each section is not clearly stated at the beginning of each section, and a reader may feel lost figuring out where the content is headed. Section 3 in particular needs to be improved by adding an overall picture and organization to it. Currently, it is a collection of disjoint topics without a coherent theme gluing them together. The supplementary material also has very little organization which makes it difficult to verify the result; adding cross reference between main and supplementary sections may help in grasping the proof.”* — We have added some text at the beginning of each subsection of Section 3, to signal where the construction is going and explain how the parts connect with one another to give the complete construction and proof. Cross references in the main section point to the appropriate part of the supplementary material for additional technical detail. Additional introductory paragraphs have been added to the different sections of the supplementary material to explain how these link together.
3. *“Thirdly, since this paper is based on several previous developments, many essential tools/concepts are borrowed from those previous works. In my opinion, more attempt is needed to explain what these tools are at least on a high-level. As a reader, I expect to follow the details, at least on a conceptual level, by reading the paper by itself and consult the previous works for further details. My impression is that the paper may not*

be accessible to an audience who has not studied to the previous CPGW and BCLPG constructions closely. Lastly, there are several vague sentences and simple errors across the text.” — We have attempted to clarify what the separate steps of the proof are in Section 3 by removing some of the mathematical details in the proof sections and giving some additional overview in the sections to give more high-level intuition into how the result works. Furthermore, in section 3.1 we have added an overview of techniques used in previous results and where our results and techniques differ from those.

Individual Comments:

1. *Line 278: It is impossible to encoding* → *It is impossible to encode* — typo corrected to “impossible to encode”.
2. *Line 118: ‘axiomatic and algorithmic sense’ too vague.* — this sentence has been removed and replaced by talking about the Halting Problem instead.
3. *“Different pieces in Section 3.2 seem disjoint. It would be easier for the reader to follow if you first give a coherent picture about how these pieces are going to be connected. For example, starting the section it is not clear whether the section is sketching the proof or just presenting disjoint pieces to be put together later.”* — we added a brief non-technical preamble at the beginning of the section “The Encoded Computation”. This should make it clear how the following parts are related.
4. *“3.2 Phase estimation: I assume . . . eigenvector etc.”* — added the statement: “the algorithm takes as input the eigenvector corresponding to $e^{i\pi\varphi}$ and outputs . . .”.
5. *“In Section 3.2: ‘interactions depending on phi and not |phi|’ is vague . . . phi?”* — this part of the section has been deleted to simplify the section as a whole.
6. *“It would be worth mentioning in 3.4 that the Wang tiling problem is undecidable.”* — We have not made this change: although the reviewer is correct in their statement, we are worried that this remark may confuse readers and that they may misread this to mean that the undecidability of our result is fundamentally due to the undecidability of the Wang tiling problem (which is not true).
7. *“Line 263: ‘.. is, cannot be ..’ probably a typo?”* — corrected to “cannot be”.
8. *“Line 271: stability in perturbation of phi is not mentioned in the main text but highlighted in the discussions section.”* — we have added a small paragraph towards

the end of the discussion section to include what we mean by stability (that the properties of the Hamiltonian are preserved under a small perturbation). How this corresponds to the discussion in the main text should now be evident.

9. “*Hard to follow what the purpose of Section A is. E.g., I would add more introductory remarks before the content presented in A1.*” — a small introduction has been added before the term section “A.1. The State of the Art”. There is also a minor sentence change in the section “A.1. The State of the Art”.
10. “*Line 415; is definition R_n correct?*” — corrected to “ $\mathbf{R}_n = \begin{pmatrix} 1 & 0 \\ 0 & 2^{i\pi 2^{-|\varphi|}} \end{pmatrix}$ ”.
11. “*I don’t follow 425 and eq above. Also eqn right below 445.*” — equation above line 425 was not correct and required β_0 to be changed to $|\beta_0|$. We have explained the set of equations more clearly. The same correction was needed for the equation below line 445.
12. “*1233 is not a sentence*” — sentence was removed and the paragraph labelled “More Realistic Systems” has minor rewrites.
13. “*Line 39 may have a misplaced comma.*” — sentence has been rewritten with comma removed.
14. “*Line 47: does \rightarrow does not?*” — correct as per reviewer’s suggestion.
15. “*Is superconductivity an example a phenomenon related to the characterization of the phase of matter discussed in this paper. My impression is that this paper is about zero temperature phase transitions.*” — we have removed references to superconductivity.
16. “*Line 26. ‘;’ \rightarrow ‘:’*” — changed as per recommendation.
17. “*Line 47: ‘logically impossible’ is somewhat vague*” — this part of the sentence has been removed and this entire section of the paper has been reformed and rewritten.
18. “*Line 110: ‘there exists a halting instance eta ..’ is too vague. What does it mean for eta being a halting instance of M? The wording can be clarified to include the construction interprets phi as a string of bits representing the description of some TM relative to some universal TM.*” — we have modified the statement of the theorem to make it clear that a halting instance is one for which the TM eventually halts, and that the UTM interprets the input as an encoding of a TM.

19. “*Before Corollary 26 it would make sense to describe what a phase diagram is in more precise terms.*” — a small sentence is added to the introduction which defines a phase diagram for a k -parameter Hamiltonian, and how the phase can be related to (or defined by) some observable.
20. “*Section 3.1 has no introductory sentences. It is not clear what is going to happen.*” — we have added introductory sections to section 3 as a whole. In particular we have added a new subsection “3.1 Overview” explaining the general proof method and how it differs from previous, similar results.
21. “*Line 140: It would help to say a sentence or so clarifying what QTM is. From what it can be perceived, the way QTM is used is as if it is quantum circuit. E.g. what if the QTM does not halt? The state in equation (3) does not make sense.*” — This was addressed by removing line 140 to avoid confusion and replacing it with a brief description of what a QTM can be thought of at the beginning of Section 3.2. We have note that equation (3) does not necessarily encode the computation until the QTM halts, rather the number of steps T which are encoded is dependent on the circuit-to-Hamiltonian mapping and the Hamiltonian’s size, not on whether the computation halts. A line has been added in to address this.
22. “*The paragraph after line 158: you explain QPE and at the end you end up with a quantum state $|\chi(\varphi)\rangle$. But what U_φ is used here? Plus the input of a QPE must include an eigenvector.*” — have now included a defintion of $U_\varphi = \begin{pmatrix} 1 & 0 \\ 0 & e^{i\pi\varphi} \end{pmatrix}$.
23. “*Section 3.3 is not possible to follow.*” — The section “Tiling and Classical Computation” has been completely rewritten. More motivation for the tiling Hamiltonian is included and we try to give a higher level overview. Some of the mathematical details have been removed.
24. “*Line 189: ‘We now want to combine classical and quantum Hamiltonians together.’ Which classical and quantum Hamiltonians? ”* — this was modified to the statement: “We now want to combine classical Hamiltonian encoding the Wang tiles, and Feynman-Kitaev quantum Hamiltonian together in a particular way.”
25. “*Line 271: I don’t recall any discussion about stability in phi before this sentence.*” — see point 8.

26. “Line 132: “the TM”. Which TM?” — we have completely rewritten this section. Section 3 now has a short introductory paragraph and a new section called “Overview and Previous Literature”.
27. “Line 735: ‘proof’ → prove” — changed as per the author’s request. We also fixed a typo in the same sentence where it should read “. . . rigorously how the highest net-bonus tiling **is achieved**, we break the proof . . .”, where the correction is emphasised in bold.

Reviewer 2 Corrections

1. “I should also say that I am somewhat confused by the very statement of the main corollary, Corollary 2.5. What exactly is meant by ‘the phase of $H(\varphi)$ is uncomputable’? In fact, what is ‘the phase of $H(\varphi)$ ’? Since this is in fact the central claim of the paper, it should also be perfectly clear.” — Have rewritten Corollary 2.5 to make it clear that the Hamiltonian is in one of two phases for any given value of the parameter, and determining which phase it is in for this value of the parameter is undecidable.

Reviewer 3 Corrections

Major Comments:

1. “My main complaint is that the notion of phase diagram is used too broadly in the title, abstract, and introduction and this can easily to misunderstandings about what is really shown here. While the abstract references high-temperature superconductivity (a seemingly random example), the precise meaning of the term ‘phase diagram’ in the present context is only given later, in Corollary 2.5. To be clear, most physicists define a phase transition “as a discontinuous change of a macroscopic observable (like total magnetization or one of its derivatives) or as a change of a discrete topological index. While high-temperature superconductivity falls into this category, it is less common to define a quantum phase” directly in terms of the existence or absence of a gap. (Of course, the closing of a bulk spectral gap indicates the possibility of a change in the topological behavior of the ground state, but (a) the topological property is really what defines the nature of the phase, not the gap, and (b) the gaplessness in that scenario occurs at the phase transition point which is not usually thought of as a separate phase.) This is all to say that the authors prove uncomputability of the gapped

ver- sus gapless phase diagram". While this is a diagram showing different types of correlation behavior in the system (and thus a phase diagram in a much looser sense of the word), it is not a phase diagram in the other sense described above and the abstract and introduction should make this fact much more clear than they presently do." — We agree with the reviewer that our definition of phase was confusingly formulated. We believe that our result does coincide with the notion of phase specified by the reviewer, though we agree this was not at all clear from our previous presentation. We have significantly rewritten this part of the exposition to try to make this much clearer. In particular, we have changed the way the result is stated to reflect this, by showing one can define an observable which is either 0 or 1 on the separate phases, which provides an macroscopic order parameter distinguishing the two phases, and that there is a discontinuous change between the two which coincides with the closing of the spectral gap. One of these phases corresponds to a gapped phase with trivial correlations, while the other is a gapless phase with algebraic decay of correlations. We highlight that the gapped/gaplessness is not what defines the phase, but is merely a feature of them since in this particular system one phase is a gapped, the other is a gapless phase. This allows additional implications beyond uncomputability of the phase diagram, but is not the key result. However, the phases themselves are identified by a global order parameter. We hope that the revised presentation makes this much clearer, and thank the referee for prompting us to improve this aspect.

We further add that if X is some property of a Hamiltonian in the thermodynamic limit, and let h_X and $h_{\neg X}$ be two sets of local terms which do and do not have the property X in the thermodynamic limit. Then with a simple modification, we can change the Hamiltonian by removing the $|0\rangle\langle 0|$ and h_d parts of the local terms and replacing them with h_X and $h_{\neg X}$ terms, and again obtain uncomputability of the phase diagram. However, now the two phases have the property " X " or " $\neg X$ ", which can be chosen to be almost any desired non-trivial property. This is a trivial modification to the construction, but highlights the strength of the uncomputability implications, as it shows that uncomputability obtains regardless of the choice of order parameter or any other feature distinguishing two phases. We have added a paragraph in the "Discussion and Implications" section to clarify this additional point.

2. *"A second complaint I have is that the authors should probably not sell the implications of their result on the empirical method of studying larger and larger volumes to extrapolate to thermodynamic behavior as strongly as they do. The constructed Hamiltonian is highly artificial and the proven undecidability result almost certainly*

has no bearing on the Hamiltonians that are actually studied in this way. In fact, for frustration-free spin systems, such as AKLT models, there even exist rigorous finite-size criteria for spectral gaps in two-dimensions which depend continuously on underlying parameters and can thus be used to identify gaps for a range of parameter values. While these methods have their limitations, if one comes across a random model in research it is far more likely that these general methods apply and one can infer rigorous information from finite-volume behavior than that the gapped versus gapless phase diagram is uncomputable. I actually think that the authors, all very respectable researchers, agree with me on these points and I would just suggest that they explain them more clearly.” — We indeed completely agree with the reviewer on this point, and have added some text to the end of the first section of the “Discussion and Implications” to emphasise this important point: “We note, however, that our result only applies to frustrated Hamiltonians, and that for many commonly occurring Hamiltonians — particularly those with small local Hilbert space dimension — determining the phase will often be decidable.”

Minor Comments and Corrections

1. *“The convention that φ denotes the length of the binary expansion of φ should be stated when $|\varphi|$ first appears on page 2, whereas right now it is first stated in Section 4.”* — changed so that $|\varphi|$ is defined when it first appears.
2. *“At the top of page 7, the fundamental and pioneering nature of [4] could be made more clear than it is right now.”* — due to word-count constraints we weren't able to make a note of this on page 7, but have instead opted to write a paragraph in the supplementary material “Section C QPE and Universal QTM Hamiltonian” which explains the importance of the Gottesman and Irani result.
3. *“The footnote on page 10 has a typo: length of the” is written twice.”* — duplication was removed.
4. *“Even if it is very standard, it should be noted once that QFT stands for Quantum Fourier Transform.”* — in the section “A.1. State of the Art”, at the first occurrence of the term “quantum Fourier transform” a bracket term “(QFT)” has been added.
5. *“It would be of benefit to the wider community if the authors could do a better job at comparing their techniques and ideas to the ones in the previous works [1, 3]. Given the overlap of authors between these works, this is presumably easy to do and I believe*

it would help others to identify the key methodological differences. Only the section on quantum phase estimation is fully clear in this regard. As a concrete example, it would be helpful to explain how the tiling and marker Hamiltonian used here differ from the implementation via the quasiperiodic Robinson tiling in [3].” — We have a preamble to section 3, and an overview in subsection 3.1 which hopefully both clarify the overall structure and specify how our methods differ from those of the previous works.

Other Corrections

1. Line 233 “high” was changed to “highly”.
2. Section 3.1 (now Section 3.2), we have redefined the history state such that we no longer make reference to the unitaries encoding the evolution of the QTM. Instead we just define it in terms of the state after t steps of the computation.
3. In the section “Tiling and Classical Computation” we have removed references to the point at which the \bullet marker is placed: we only mention that it is determined by the action of the TM.

REVIEWERS' COMMENTS

Reviewer #1 (Remarks to the Author):

Thanks for the edits. The changes make a significant change in explaining the result and communicating the flow of the paper. However, some of these new sentences might need slight improvements and edits. Furthermore, part of my concerns from before specifically about section 3 still persist. Let me try to express some of them in the following. Please note the line numberings are based on the manuscript which highlights the added/removed sentences.

Several comments for Section 3:

1. The section starts with saying we will sketch each step of the proof. The section then starts with an overview subsection. I expected to see a recipe/ summary of how to put together the following sections 3.2, 3.3 and 3.4. etc. However the section only contains a general explanation of the previous constructions and ends by saying that they fall short here. The subsections after that (ie 3.2, 3.3, ...) still seem disjoint not following a predefined trend. In particular, the way the title of Section 3 suggests, I expected to see the overview description of a final output Hamiltonian with missing pieces (ie H_{comp} , H_{cb} , H_{grid}) to be explain the follow up sections.
2. Section 3.3 is titled the encoded computation. How is this encoded computation used in the final construction? The description of the dovetailed computation is vague. In addition the way the title suggests, I expected 3.3 to start by outlining the description of an encoded computation with missing pieces and I expected the subsections of 3.3 to explain those missing pieces. However, the subsections does not seem to serve such or any other prescribed roles.
3. In 3.3., the subsections explain that the encoded computation is by taking a unitary and a direction and outputting a state containing the result of a phase estimation then feeding that state to a universal QTM then mapping that universal QTM on the particular input to a Hamiltonian. Where is this input unitary coming from in the proof of the main result?
4. In Section 3.4 there is a mention of ϵ being large or small. I do not recall this being discussed in 3.3.. What is the situation with intermediate values of ϵ ? What is the role of boosting a gap? (I believe this is expressed later through equation 7). The way I understood the paper, the motivation behind this gap amplification deserves more explanation/outline because from other sections seems to be an important construction in this paper. What is the motivation behind Wang tiles? Why do we need the expressed properties of the classical tiles for the gap amplification?
5. Paragraph (phase estimation) right after line 218: is imperfect QPE an artifact of a continuous ensemble?

There some more comments which I include in the following.

1. Line 11. As a minor point, saying ϕ in R may be problematic when claiming a positive measure of parameters are undecidable instances.
2. Line 75: invariant under τ \rightarrow invariant under σ
3. The paragh starting line 71 emphasizes that gapped vs gaplessness is secondary and there is a macroscopic observable $O_{A/B}$ whose estimation is not computable. However no mention of $O_{A/B}$ is given in the results section or other sections. Also is this observable efficiently measureable? Meaning, hypothetically, if one builds an infinitely large material based on this construction, is there a simple experiment to distinguish phases A from B?
4. Line 154: saying that there is no deterministic algorithm may imply that there could be a randomized one etc. Line 152: wouldn't undecidability in algorithmic sense also imply undecidability in axiomatic sense? In particulare I believe this paper directly proves the former.
5. Why is boosting the gap required here but not the previous work?
6. In line 186 where you say "however this is not enough for the uncomputability of phase diagrams.." is a great place to add a (if possible) short reason for why.
7. eqn 3: replace ψ_i by ψ_t ? What if the TM does not halt? Should it be added that the Hamiltonian simulates the QTM for T steps, for predefined and fixed T . Otherwise this equation is not well-defined for a non-halting TM.
8. The use of word "meaning" in line 182 may imply that undecidability of a spectral gap is immediatly implied from the undecidability of the HALTING problem
9. Definition of a QTM is missing
10. Line 364: should "necessary \rightarrow sufficient"?
11. Line 216: After the first QTM has run ... : I am not sure what the "the first and second QTM" are referenced from.
12. The added comparison of techniques with previous works of section 4 is nice. There it is highlighted that there are two main technical improvements added to previous work. (1) dealing with approximate QPE and (2) self adjusting length for the encoding. Why is (1) a bottleneck in this work but not the previous work? Why (2) is a necessity in this work but not the previous?

Reviewer #3 (Remarks to the Author):

My comments have been fully addressed in the revision. In particular, I am pleased to see the overview in Section 3.1 and the connection to expectation values of macroscopic observables clarified.

I recommend the paper for publication.

Authors' Response: Uncomputability of Phase Diagrams

Johannes Bausch^{*1}, Toby S. Cubitt², and James D. Watson^{†2}

¹CQIF, DAMTP, University of Cambridge, UK

²Department of Computer Science, University College London, UK

November 13, 2020

Summary In the first section we address the reviewer's comments. In the "Other Changes" section we go through the diff file and explain the changes made.

Reviewer 1 Corrections

Broad Comments:

“Several comments for Section 3: The section starts with saying we will sketch each step of the proof. The section then starts with an overview subsection. I expected to see a recipe/ summary of how to put together the following sections 3.2, 3.3 and 3.4. etc. However the section only contains a general explanation of the previous constructions and ends by saying that they fall short here. The subsections after that (ie 3.2, 3.3, . . .) still seem disjoint not following a predefined trend. In particular, the way the title of Section 3 suggests, I expected to see the overview description of a final output Hamiltonian with missing pieces (ie H_{comp} , H_{cb} , H_{grid}) to be explain the follow up sections.”

*Email: jkrb2@cam.ac.uk

†Email: ucapjdj@ucl.ac.uk

We have expanded this section so that it now details the previous works in more depth, and gives a more extensive overview of how the separate parts of the construction combine together. We also now link each part of the description to the section where more details are given, outline which parts of the new construction have a significant novel contribution, and compare these to the previous results.

“Section 3.3 is titled the encoded computation. How is this encoded computation used in the final construction? The description of the dovetailed computation is vague. In addition the way the title suggests, I expected 3.3 to start by outlining the description of an encoded computation with missing pieces and I expected the subsections of 3.3 to explain those missing pieces. However, the subsections does not seem to serve such or any other prescribed roles.”

We have modified the introduction to the section “The Encoded Computation” in order to clarify this point, and expanded the explanation in this section. It now outlines that there are two QTM’s which run in succession: the first runs phase estimation, the second then takes the output of the first as input and then runs a universal TM. We then outline how the halting property relates to the ground state energy when this is encoded in the Hamiltonian. The subsections then outline the action of each QTM’s separately in more detail.

“In 3.3., the subsections explain that the encoded computation is by taking a unitary and a direction and outputting a state containing the result of a phase estimation then feeding that state to a universal QTM then mapping that universal QTM on the particular input to a Hamiltonian. Where is this input unitary coming from in the proof of the main result?”

We have added a brief explanation at the beginning of the section “The Encoded Computation” highlighting that the unitary U_φ is encoded in the matrix elements of the Hamiltonian.

“In Section 3.4 there is a mention of ϵ being large or small. I do not recall this being discussed in 3.3. What is the situation with intermediate values of ϵ ? What is the role of boosting a gap? (I believe this is expressed later through equation 7). The way I understood the paper, the motivation behind this gap amplification deserves more explanation/outline because from other sections seems to be an important construction in this paper. What is the motivation behind Wang tiles? Why do we need the expressed properties of the classical tiles for the gap amplification?”

We have changed the phrasing of this and now explicitly cover all cases for ϵ . Furthermore, we've added a brief explanation detailing what happens even in the case of ϵ not close to 1 or 0. We have also included additional explanation and referenced previous literature on the gap amplification the beginning of the methods section.

“Paragraph (phase estimation) right after line 218: is imperfect QPE an artifact of a continuous ensemble?”

This is (in part) the case. We felt it may be confusing to add in a discussion about this at line 218, so have instead added a discussion of why approximate QPE is needed in first part of the “Methods” section.

Individual Comments:

“Line 11. As a minor point, saying phi in \mathbb{R} may be problematic when claiming a positive measure of parameters are undecidable instances.”

This is a subtle and important point, and we thank the reviewer for prompting us to explain this better. We have tightened the wording of the main corollary to avoid potential confusion on this point, and we have added a new subsection to the supplementary material titled “Subtleties Concerning Computable and Uncomputable Numbers” which discusses the mathematical subtleties of the result concerning undecidability over the reals in more detail.

“Line 75: invariant under to \rightarrow invariant under”

This sentence has been removed.

“The paragh starting line 71 emphasizes that gapped vs gapplessness is secondary and there is a macroscopic observable $O_{A/B}$ whose estimation is not computable. However no mention of $O_{A/B}$ is given in the results section or other sections. Also is this observable efficiently measureable? Meaning, hypothetically, if one builds an infinitely large material based on this construction, is there a simple experiment to distinguish phases A from B?”

We have edited this so that $O_{A/B}$ now appears in the main results section, and it is explained there that computing its ground state expectation is as hard as solving the Halting Problem. We have also added a brief explanation that if we restrict this observable to some part of the lattice, it will have the same expectation as the full observable, thus in principle allowing the phase to be identified by finite, local measurements.

“Line 154: saying that there is no deterministic algorithm may imply that there could be a randomized one etc. Line 152: wouldn’t undecidability in algorithmic sense also imply undecidability in axiomatic sense? In particular I believe this paper directly proves the former.”

We agree with the referee that this was poorly worded, and misleading as the referee has noted. The word “deterministic” has been deleted. Furthermore, we have added a brief discussion of axiomatic undecidability in the Discussion section.

“Why is boosting the gap required here but not the previous work?”

This was not sufficiently well explained in the previous submission, which gave the referee the wrong impression on this point. Boosting the gap *is* required in previous works, but the methods used previously are insufficient to give the uncomputability of phase diagrams results described in this paper. We have added a fuller discussion of this in the overview of previous works, in the numbered points towards the beginning of the Methods section.

“In line 186 where you say “however this is not enough for the uncomputability of phase diagrams..” is a great place to add a (if possible) short reason for why.”

This has been rewritten to explain why this is not the case, and this section is now part of the first part of the Methods section.

“eqn 3: replace ψ_i by ψ_t ? What if the TM does not halt? Should it be added that the Hamiltonian simulates the QTM for T steps, for predefined and fixed T . Otherwise this equation is not well-defined for a non-halting TM.”

ψ_i has been changed to ψ_t . We have added a sentence explaining that T is a fixed function of the Hamiltonian’s size, and is determined by the particular form of the QTM-to-Hamiltonian mapping.

“The use of word “meaning” in line 182 may imply that undecidability of a spectral gap is immediately implied from the undecidability of the HALTING problem”

We have rewritten this line and the line before it to clarify this.

“Definition of a QTM is missing”

A concise explanation of a QTM is now given at the beginning of section “Encoding Computation in Hamiltonians”.

“Line 364: should “necessary → sufficient?””

This line has now been deleted.

“Line 216: After the first QTM has run . . . : I am not sure what the “the first and second QTM” are referenced from.”

This has been changed to add that the first QTM runs a QPE protocol and the second is universal Turing Machine.

“The added comparison of techniques with previous works of section 4 is nice. There it is highlighted that there are two main technical improvements added to previous work. (1) dealing with approximate QPE and (2) self adjusting length for the encoding. Why is (1) a bottleneck in this work but not the previous work? Why (2) is a necessity in this work but not the previous?”

The section describing the techniques of previous results now appears toward the beginning of the Methods section, and this now outlines why these steps are necessary and how they relate to previous work.

Other Changes

We have made other changes to the file. For the most part, these are either minor changes made for readability or clarity, or they are changes made to satisfy the requests in the author checklist supplied by the editor.

All line references below are given as per the diff file.

Abstract Minor changes to wording for clarity.

Introduction: Lines 1-54 Rearranged sentence at lines 35-37 to appear at more appropriate place at lines 42-43. Other minor changes are polishing.

Introduction: Lines 55-93 These lines have been removed from the introduction and moved to the Discussion section to comply with the request in the author checklist. The Introduction section now contains a concise summary of the current work rather than the more extended discussion that existed before.

Introduction: Lines 94-103 Changed sentence for readability.

Introduction: Lines 104-128 The sections that have been removed have been moved to the Discussion section later. The parts that remain and appear to have been added are just sections of the old introduction that have been rearranged to comply with author checklist request of a more concise summary of the work.

Results: Lines 129-148 Minor rewrites for clarity. Added in references to supplementary information.

Results: Lines 149-186 Small changes to theorem and corollary statements have been made to address referee's concerns about the parameter being defined over the reals. Other small changes outside of these lemmas have been made for readability purposes.

Discussion: Lines 186-234 The entire discussion section was brought forward in the text to bring the article in line with Nature Communications formatting specifications. The lines 186-234 were previously part of the introduction and have been moved to the discussion as per the formatting instructions in the author checklist concerning the "Main Text".

Discussion: Lines 235 - 316 There is now a slightly longer discussion of the further research directions, as requested in the author checklist. Other than these, there are no major changes other than those already discussed in the response to the referees (such as the discussion of axiomatic independence). There are minor rewordings for clarity and the section discussing the implications with respect to numerical results has minor changes to emphasise previous work.

Methods: Lines 317-434 This part of the Methods section has been edited to address the concerns of the first comment from the referee. Small changes to the text have also been made for readability.

Methods: Lines 435-498 These changes have been made to satisfy the referee as per the second and third request. Also the definition of the QTM has been added, as per the referee request.

Methods: Lines 499-526 Small changes here have been made to address the referee's comments about the value of ϵ . Other than some minor rewordings, there are no other changes.

Methods: Lines 527-601 Small changes have been made to improve readability and to bring the article within Nature Communication's editorial guidelines (e.g. by removing quotation marks).

Methods: Lines 602-635 We have added some additional maths taken from (what was previously) Section F to clarify the statements about the ground state energy and the spectrum of the Hamiltonian. This was done to address the relevant point in the author checklist in the "Methods and Data" section.

Methods: Lines 636-646 Minor changes and polishing for readability

Lines: 647-717 The discussion section has been moved so that it now appears before the methods section, as per the author checklist.

Lines: 729-730 Added competing interests statement.